# The Effects of Polyphenols on Bone Metabolism in Postmenopausal Women: Systematic Review and Meta-Analysis of Randomized Control Trials

**DOI:** 10.3390/antiox12101830

**Published:** 2023-10-05

**Authors:** Gianmaria Salvio, Alessandro Ciarloni, Claudio Gianfelice, Francesca Lacchè, Sofia Sabatelli, Gilberta Giacchetti, Giancarlo Balercia

**Affiliations:** Endocrinology Clinic, Department of Clinical and Molecular Sciences, Polytechnic University of Marche, 60126 Ancona, Italy; g.salvio@pm.univpm.it (G.S.); ale.ciarloni93@gmail.com (A.C.); claudio.gianfelice@ospedaliriuniti.marche.it (C.G.); francesca.lacche88@gmail.com (F.L.); sofiasabatelli.95@gmail.com (S.S.); gilberta.giacchetti@ospedaliriuniti.marche.it (G.G.)

**Keywords:** flavonoids, soy, osteoporosis, osteopenia, menopause, oxidative stress

## Abstract

Osteoporosis is a condition favored by the postmenopausal decline in estrogen levels and worsened by oxidative stress (OS). Polyphenols are natural compounds abundantly found in fruits and vegetables, and they exert antioxidant and hormonal effects that could be useful in osteoporosis prevention, as suggested by epidemiological studies showing a lower incidence of fractures in individuals consuming polyphenol-rich diets. The aim of our meta-analysis is to evaluate the effects of polyphenols on bone mineral density (BMD, primary endpoint) and bone turnover markers (BTMs, secondary endpoint) in postmenopausal women. Twenty-one randomized control trials (RCTs) were included in our analysis after in-depth search on PubMed, EMBASE, and Scopus databases. We found that supplementation with polyphenols for 3–36 months exerted no statically significant effects on BMD measured at lumbar spine (sMD: 0.21, 95% CI [−0.08 to 0.51], *p* = 0.16), femoral neck (sMD: 0.16, 95% CI [−0.23 to 0.55], *p* = 0.42), total hip (sMD: 0.05, 95% CI [−0.14 to 0.24], *p* = 0.61), and whole body (sMD: −0.12, 95% CI [−0.42 to 0.17], *p* = 0.41). Subgroup analysis based on treatment duration showed no statistical significance, but a significant effect on lumbar BMD emerged when studies with duration of 24 months or greater were analyzed separately. On the other hand, we found a significantly slight increase in bone-specific alkaline phosphatase (BALP) levels (sMD: 1.27, 95% CI [1.13 to 1.42], *p* < 0.0001) and a decrease in pyridinoline (PD) levels (sMD: −0.58, 95% CI [−0.77 to −0.39], *p* < 0.0001). High heterogeneity among studies and unclear risk of bias in one third of the included studies emerged. A subgroup analysis showed similar effects for different duration of treatment and models of dual-energy X-ray absorptiometry (DXA) scanner. More robust evidence is needed before recommending the prescription of polyphenols in clinical practice.

## 1. Introduction

Life expectancy is becoming longer than it was in past decades, and this leads to an increased incidence of many age-related diseases like cardiovascular and oncological diseases. For what concerns age-related endocrinopathies, osteoporosis plays a main role [1] due to its high prevalence and socioeconomical costs [2]. Osteoporosis is a characterized by reduced bone mineral density (BMD), which leads to an increased risk of pathological fractures, with a World Health Organization (WHO) worldwide estimation of about 9 million fractures per year related to osteoporosis [3].

Two main different cell types can be found in bone tissue: osteoblasts, which renew bone tissue, and osteoclasts, which remove bone tissue. In physiological conditions, osteoclast and osteoblast activity is balanced to maintain a normal BMD. In many different conditions, first of all in the post-menopausal period, there is an imbalance with increased osteoclastic activity and decreased osteoblastic activity that lead to reduced BMD. Indeed, estrogens physiologically reduce osteoclastic activity and induces osteoclast apoptosis: in the first 5–7 years of the postmenopausal period, women are expected to lose about 12% of their BMD due to a rapid fall in estrogen levels [4]. Several studies have evaluated estrogen therapy in post-menopausal women affected by osteoporosis, but the possible cancer risk in tissues that are rich in estrogen receptors like breasts, ovaries, and endometrium has always limited that use in the clinical practice [5]. Indeed, several meta-analyses showed an ovarian cancer relative risk (RR) ranging from 1.19 to 1.46 after estrogen therapy, and even when treatment lasted less than 5 years, the RR was 1.43 [6,7,8,9]. Recently, a role of oxidative stress (OS) in the development of osteoporosis has emerged. OS can increase osteoclastogenesis and decrease osteoblastic differentiation and activity while increasing osteoblastic and osteocytic apoptosis. In addition, it has been hypothesized that menopause-related decline in estrogen levels could increase bone susceptibility to OS, increasing the risk of postmenopausal osteoporosis [10].

In recent decades, many drugs have been developed to treat osteoporosis by implementing new bone formation or reducing bone resorption, leading to a significant reduction in fracture risk, despite sometimes being expensive, with potential side effects and a limited treatment time [11]. Usually, this treatment is started after an osteoporosis diagnosis, rather than for its prevention. Due to the above-mentioned therapeutic limits, efforts have been made to find a therapeutic option with fewer side effects, usable for a long time and not only able to treat but also to prevent osteoporosis [12].

Abundantly found in fruits and vegetables, polyphenols are natural compounds with a powerful antioxidant effect and can be divided into flavonoid (e.g., isoflavones such as ginestein and dadzein, icariin, and quercetin) and non-flavonoid compounds (e.g., resveratrol and curcumin) [13]. They can be extracted from flowers, bark, roots, and leaves and are particularly abundant in berries, tea leaves, and legumes such as soy and red clover [14]. Epidemiologic data suggest that dietary flavonoid intake could be related to reduced fracture risk [13]. In this purpose, a recent meta-analysis showed that a “Healthy” dietary pattern, characterized by high content in fruits and vegetables, fish, poultry, and whole grains, was associated with reduced risk of fractures and low BMD (OR: 0.65, *p* = 0.037 and 0.82, *p* = 0.028, respectively), whereas the “Meat/Western” dietary pattern was associated with higher risk of both fractures (OR: 1.11, *p* < 0.001) and low BMD (OR: 1.22, *p* = 0.028) [15] Polyphenols suppress osteoclast differentiation and activity, whereas enhance osteoblast activity in vitro [16], thus potentially counteracting the loss of BMD that is observed with advancing age and declining estradiol levels. In addition, the isoflavones have a high affinity for the estrogen receptor β (ER β) [17], which is expressed in tissues with estrogen-dependent trophism, such as bone tissue, but not for the estrogen receptor α (ER α), associated with cancerogenic effects [18,19]. Moreover, isoflavones can modulate the activity of hepatic cytochrome P450 enzyme, leading to a better 2-hydroxyestrone/16-hydroxyestrone ratio [20,21] and reducing levels of 16-hydroxyestrone, which is the more genotoxic and associated with higher cancer risk [22]. They also exert a positive effect on endothelial function [23], improving intraosseous blood flow that is important for the differentiation of stem cells in osteoblasts to obtain a new bone formation [24].

Due to these peculiar characteristics, polyphenols have been studied as an alternative therapeutic option for prevention and treatment of postmenopausal osteoporosis. Therefore, in this review, we are analyzing different studies that evaluated polyphenols treatment of postmenopausal bone loss.

## 2. Materials and Methods

This study was conducted following the guidelines of The Preferred Reporting Items for Systematic reviews and Meta-Analyses (PRISMA) statement [25]. The research was registered on PROSPERO (https://www.crd.york.ac.uk/prospero/, accessed on 28 June 2023) with number CRD42023437428.

### 2.1. Search Strategy

A systematic search was conducted through Scopus, PubMed, and EMBASE databases from May to July 2023. The terms “flavonoids”, “polyphenols”, “natural compounds”, “isoflavones”, or “antioxidants” were combined with “postmenopausal” and “osteoporosis” or “bone loss”. In particular, the following query strings were used: “flavonoids OR polyphenols OR natural compounds OR isoflavones OR antioxidants AND postmenopausal AND osteoporosis OR bone loss” (PubMed), “(TITLE-ABS-KEY (flavonoids) OR TITLE-ABS-KEY (polyphenols) OR TITLE-ABS-KEY (natural compounds) OR TITLE-ABS-KEY (isoflavones) OR TITLE-ABS-KEY (antioxidant*)) AND (TITLE-ABS-KEY (osteoporosis) AND TITLE-ABS-KEY (postmenopausal) OR AND (TITLE-ABS-KEY (bone loss) AND (LIMIT-TO (DOCTYPE, “ar”)” (Scopus), and “((flavonoids or polyphenols OR natural compounds OR isoflavones OR antioxidant*) and (osteoporosis AND postmenopausal OR bone loss)).mp.”. The search was conducted independently by two authors (CG and SS), and disagreements were resolved through discussion with a third author (GS).

### 2.2. Selection Criteria

Eligible studies were selected following the PICO model: Population (P, postmenopausal women), Intervention (I, supplementation with polyphenols alone or in combination with other routine drugs), Comparison (C, placebo or other control treatment), Outcome (O, BMD after treatment). Only randomized controlled trials (RCTs) were considered eligible. Studies in which hormone replacement or drugs affecting bone metabolism (e.g., bisphosphonates) were taken along with polyphenols were excluded. If polyphenols were administered with additional supplements (e.g., calcium carbonate or vitamin D), studies were included only if the control group received the same supplement with or without placebo. Primary outcomes were spine (L1–L4 or L2–L4), femoral neck, total hip, and whole-body BMD measured via dual-energy X-ray absorptiometry (DXA). Secondary outcomes were bone turnover markers (BTMs), including bone-specific alkaline phosphatase (BALP), C-terminal telopeptide of type I collagen (CTX), N-terminal telopeptide of type I collagen (NTX), osteocalcin (OC), procollagen I N-terminal propeptide (PINP), pyridinoline (PD), and deoxypyridinoline (DPD). Randomized control trials with parallel or crossover designs were included.

### 2.3. Data Extraction and Quality Assessment

Three authors (FL, CG, and SS) performed data extraction, which was verified by a fourth author (GS). The following data were collected: first author, year, country, study design (parallel or crossover), type of polyphenols, daily dose, route of administration, additional treatment/supplement, type of control, study duration, model of DXA scanner, number of subjects, age, BMD (g/cm^2^), and BTMs levels. Although both endpoint data and change data could be used for meta-analyses [26], a high percentage of discrepancies has been recently reported according to the initial choice of mean difference estimates [27], so we chose to include only endpoint data to produce more conservative results. When standard error of the mean (SEM) was reported, standard deviation (SD) was calculated by multiplying SEM by the square root of the number of subjects. The quality of evidence (QoE) was assessed using the Version 2 of the Cochrane tool for assessing risk of bias in randomized trials (RoB 2) [28] based on the following criteria: (1) bias arising from the randomization process, (2) bias due to deviation from intended interventions, (3) bias due to missing outcome data, (4) bias in measurement of the outcome, and (5) bias in selection of the reported results. Two researchers (FL and SS) performed the QoE, and a third author (GS) verified their entries and expressed an overall risk-of-bias judgment.

### 2.4. Statystical Analysis

The analysis was performed using RevMan software v. 5.4.1 (Cochrane Collaboration, Oxford, UK) and Comprehensive Meta-Analysis v. 4 (Biostat Inc., Englewood, NJ, USA). Standardized mean difference (sMD) and 95% confidence intervals (CIs) were calculated to compare outcome measures after treatment. I^2^ statistic was applied to inspect heterogeneity, with I^2^ > 50% and *p* < 0.1 indicating high between-study heterogeneity. If significant heterogeneity emerged, meta-analysis was performed using a random-effects model. Otherwise, a fixed-effects model was used. Studies including different group of treatment with polyphenols (e.g., different daily doses) were independently entered. For primary outcome, if more than one longitudinal measurement was available, only the one with the longest follow-up time was included. Publication bias was assessed via funnel plot asymmetry as well as Egger’s test. To investigate the source of heterogeneity, subgroup analysis (based on duration of treatment and model of DXA scanner) and sensitivity analysis (omitting each single study to explore its effect on the overall meta-analysis) were conducted. Statistical significance was set at 0.05.

## 3. Results

### 3.1. Study Selection

Using the above-mentioned search strategy, 2231 abstracts were extracted. After the removal of 702 duplicates, 1529 articles were screened. Of these, 1399 were identified by title or abstracts as papers on other topics, review articles, editorials, case reports, or animal/in vitro studies. Of the remaining 130 full-text articles assessed for eligibility, 21 [29,30,31,32,33,34,35,36,37,38,39,40,41,42,43,44,45,46,47,48,49] were included in the present meta-analysis (Figure 1). All the included studies were RCTs with parallel design, except for one [39] that had a crossover design. The main characteristics of the included studies are presented in Table 1 and Table 2. PRISMA 2020 Checklist can be found in Appendix A.

### 3.2. Risk of Bias Assessment

Risk of bias assessment revealed high risk of bias in seven studies (33.3%) [29,33,34,39,42,46,47], uncertain risk in seven studies (33.3%) [30,32,35,38,40,43,49], and low risk of bias in the remaining seven studies (33.3%) [31,36,37,41,44,45,48]. Details of the evaluation are shown in Figure 2. 

### 3.3. Primary Outcome: Effects of Polyphenols on Bone Mineral Density

The effects of polyphenols on BMD in different sites were evaluated separately. Eighteen studies including a total of 1711 subjects reported data regarding lumbar BMD. The administration of polyphenols led to negligible effects on lumbar BMD (sMD: 0.21, 95% CI [−0.08 to 0.51]), with high heterogeneity between studies (I^2^ = 88%) (Figure 3a). Visual examination of funnel plots suggested significant publication bias (Figure 3b), as confirmed via Egger’s test (*p* = 0.005). Sensitivity analysis revealed that the exclusion of the study by Marini et al. [38] led to a slight decrease in heterogeneity (I^2^ = 77%), without significant changes in the estimate of the effect size (sMD: 0.15, 95% CI [−0.08 to 0.39]).

The effects of polyphenols on femoral neck BMD were evaluated in 12 studies and 1443 patients. The analysis revealed no significant effects (sMD: 0.16, 95% CI [−0.23 to 0.55]), with high heterogeneity among studies (I^2^ = 92%) (Figure 4a). No significant publication bias emerged (Figure 4b, Egger’s test *p* = 0.133), whereas the exclusion of the study by Marini et al. [38] led to a slight effect on heterogeneity (I^2^ = 85%) and a moderate change in estimate of effect size, which remained unsignificant (sMD: 0.24, 95% CI [−0.08 to 0.57]).

The analysis of 10 studies (986 subjects) revealed no significant effects of polyphenols on total hip BMD (sMD: 0.05, 95% CI [−0.14 to 0.24]), with high heterogeneity among studies (I^2^ = 52%) (Figure 5a) and no significant risk of publication bias (Figure 5b, Egger’s test *p* = 0.956). Heterogeneity was low after removing the study by Gui et al. [33], and a slight insignificant change emerged in the estimate of the effect size (sMD: 0.11, 95% CI [−0.04 to 0.27], I^2^ = 25%).

Finally, seven studies evaluated the effects of polyphenols on whole-body BMD, revealing no significant effects (sMD: −0.12, 95% CI [−0.42 to 0.17]), with high heterogeneity among studies (I^2^ = 70%) (Figure 6a) and no significant risk of publication bias (Figure 6b, Egger’s test *p* = 0.201). The removal of the study by Kenny et al. [35] greatly decreased heterogeneity (I^2^ = 24%) but did not affect the results of the analysis sMD: 0.03, 95% CI [−0.17 to 0.23]).

### 3.4. Secondary Outcome: Effects of Polyphenols on Bone Turnover Markers

The bone markers reported most frequently in the studies (and those on which a meta-analysis could be performed) were serum BALP (five studies, 1018 patients), urinary DPD (four studies, 728 patients), serum OC (three studies, 484 patients), and urinary PD (two studies, 463 patients). The administration of polyphenols led to significant increase in BALP levels (sMD: 1.33, 95% CI [0.38 to 2.28]), with high heterogeneity among studies (I^2^ = 97%) (Figure 7a). Conversely, polyphenols significantly decreased PD levels (sMD: −0.67, 95% CI [−1.03 to −0.32]), with high heterogeneity among studies (I^2^ = 57%) (Figure 7d). On the other hand, no significant effects on DPD and OC levels emerged (Figure 7b,c). No significant risk of publication bias emerged.

### 3.5. Subgroup Analysis

Subgroup analysis was conducted to explore if different duration of treatment could significantly modify the effects of polyphenols on BMD. Three subgroups (study duration < 12 months, between 12 and 24 months, and ≥24 months) were considered for lumbar and femoral neck BMD, whereas two subgroups (<12 months or ≥24 months) were considered for total hip and whole-body BMD. Regarding lumbar BMD, the test of subgroup differences indicated no statistically significant subgroup effect (*p* = 0.09), suggesting that study duration did not significantly affect the effects of intervention. However, it should be noted that the improvement in BMD became significant only when study duration was 24 months at least (sMD: 1.00, 94% CI [0.19 to 1.81]), with persistently high unexplained heterogeneity (I^2^ = 58.3%) (Figure 8).

The test for subgroup differences indicated no statistically subgroup effect for femoral neck BMD (*p* = 1.00, I^2^ = 0%, Figure 9), total hip BMD (*p* = 0.30, I^2^ = 8.0%, Figure 10), and whole-body BMD (*p* = 0.83, I^2^ = 0%, Figure 11) as well, meaning that duration of treatment does not significantly influence the effects of polyphenols on BMD measured at these sites.

A subgroup analysis was also performed to assess if the observed effect size could be affected by the type of DXA scanner (GE Lunar or Hologic QDR), showing no statistically significant differences in any of the evaluated sites. 

## 4. Discussion

Polyphenols are a large group of phytochemicals that include several sub-classes, such as isoflavones (e.g., daidzein, genistein, and glycetein), flavonols (e.g., quercetin), and stilbenes (e.g., resveratrol). Isoflavones occur widely in nature in foods such as tea, cocoa, blueberries, and soybeans and have numerous beneficial effects on the human body [50], including protection from prostate cancer [51] and cardiovascular diseases [52]. In vitro studies showed that isoflavones exert a proestrogenic activity on bone, inducing osteoclast apoptosis through activation of ER and decreasing their activity by regulating the ratio between RANK ligand and osteoprotegerin, which are the main regulators of osteoclastogenesis. In addition, they stimulate osteoblast activation by promoting the bone morphogenic protein cascade, with a favorable effect on bone metabolism [53]. In addition, the antioxidant effects of polyphenols in human primary osteoblasts lead to increased expression of osteocalcin and collagen1A1, with concurrent enhanced cell viability [54]. In vivo, large epidemiological studies suggested that higher soy consumption may be associated with lower risk of hip fracture in unselected women [55] and in those with previous fractures, especially at the beginning of menopause [56]. Recently, an inverse correlation between risk of osteoporotic fracture and dietary intake of soy isoflavones was reported in men as well [57], confirming their potential role in the treatment of osteoporosis. 

The results of the present meta-analysis suggest that the polyphenols administration exerts negligible effects on BMD in women with postmenopausal osteoporosis, with high unexplained heterogeneity among studies. Notably, we found high variability in dosage and type of polyphenols, with some studies using pharmacologic formulations (e.g., tablets, capsules) and other polyphenol-enriched diets. Moreover, significant risk of bias emerged in two thirds of the included studies, especially due to unclear blinding modalities, and concrete risk of publication bias also exists for studies in which lumbar BMD was evaluated. Our findings are aligned with the recent meta-analysis by Li et al., who investigated the effects of resveratrol on BMD (lumbar, total hip, and whole body) values in 10 RCTs, reporting no statistically significant effects at any site [58]. Conversely, in a previous meta-analysis, which included 10 RTCs, Ma et al. reported that soy isoflavones administration could significantly attenuate bone loss in peri- and postmenopausal women [59]. Incidentally, it should be noted that the authors did not state if included patients suffered from osteoporosis or not, and the reported effect size (a mean increase of 0.02 g/cm^2^ in the active group versus placebo), despite being statistically significant, may not have exceeded the least significant change [60] to be considered clinically relevant. In detail, there was a similar increase in a postmenopausal woman with normal BMD (e.g., L1–L4 BMD 1.200 g/cm^2^, T-score 0), which equals to 1.7%, whereas in an osteoporotic woman with reduced BMD (e.g., L1–L4 BMD 0.890 g/cm^2^, T-score −2.5), it equals to 2.3%. In both cases, the precision error of the DXA scanner(s) should be determined to establish if a similar effect size could be considered statistically significant [61], and then clinically relevant. Similarly, in another recent meta-analysis, Akhlaghi et al. evaluated the effects of soy isoflavones in the prevention of bone loss at lumbar spine, hip, and femoral neck in 30 studies involving women and men independently according to their bone health status or menopausal state. Interestingly, the authors reported statistically significant improvement in BMD for patients in the active group compared with the control group ranging from 0.22% (at hip) to 2.27% (at femoral neck), with an intermediate improvement at lumbar spine (0.76%), but they questioned the clinical significance of their results in terms of effective fracture risk reduction [62]. For this purpose, treatment-related changes in BMD were recently validated via a large meta-regression analysis as a good surrogate biomarker for fracture risk reduction [63]. Accordingly, surrogate threshold effect (STE) for change in total hip BMD were proposed, showing a significant fracture risk reduction at vertebral, hip, and nonvertebral site for treatment-related BMD changes exceeding STE [64]. Notably, the minimum change in total hip BMD associated with fracture risk reduction was 1.4% at 24 months of treatment, which is much higher than that observed by Akhlaghi et al. in their meta-analysis (i.e., 0.38% for intervention duration ≥ 1 year) [62]. Taken together, these data do not seem to support a relevant biological effect, even in the face of a statistically significant improvement in BMD. Furthermore, there are currently no RCTs that have evaluated the actual polyphenol supplementation efficacy to effectively reduce fracture incidence.

Another important aspect is adherence to therapy. According to a recent review, the mean persistence of oral bisphosphonates for 1 year ranges from 17.7% to 74.8%, and decreases to 12.9–72.0% at 2 years [65]. Similarly, in a large cohort of osteoporotic patients, good adherence to therapy with denosumab (i.e., an average interval of denosumab injection within 7 months) was reported in only 56.9% of patients [66]. Regarding the causes of discontinuation, a recent study revealed that more than a half of patients who interrupt the treatment decide to withdraw from drugs on their own initiative, and among them, the main motivation (46.3% of cases) is forgetting about the visit at the outpatient clinics, whereas only 15.2% of patients discontinue drugs due to medication-related factors [67]. Notably, despite generally being considered safe and free from significant side effects, a high drop-out rate from polyphenols was reported in several cases, reaching 58.1% [44] due to different causes, namely personal reasons (including low palatability) and mild gastrointestinal side effects. In addition, real adherence to treatment was assessed only in a minority of studies [29,35,36,42,43,45-47]. Therefore, since adherence to antiosteoporotic therapy remains one of the key issues in the treatment of osteoporosis, the poor tolerability of polyphenols does not make them an attractive alternative.

Of note, several included studies evaluated the effects of polyphenols on BMD after only 6 [29,32,39-41,47,49] or even 3 months [42]. According to recent guidelines issued by the American Association of Clinical Endocrinologists/American College of Endocrinology (ACEE/ACE), BMD should be monitored by DXA every 1–2 years [68]. Therefore, a shorter follow-up time could have led to underestimate the effects of polyphenols on BMD, but our subgroup analysis revealed that the observed results were not affected by duration of the study, making this hypothesis questionable. 

Interestingly, we observed that the administration on polyphenols could lead to a favorable increase in bone deposition markers (namely BALP) and a decrease in bone resorption markers (PD). This trend was already suggested by a previous meta-analysis that was focused on soy isoflavones, in which changes in BTM levels were considered as the primary outpoint [69] but lacked statistical significance. In line with these results, the administration of resveratrol did not result in significant changes in BMTs in the recent meta-analysis by Li et al. [58]. 

Taken together, current findings from RCTs seem to fully confirm the epidemiological evidence in terms of bone health. Since cross-sectional studies do not accurately investigate the cause–effect relationship, it is possible that additional factors go into delineating the relationship between polyphenol intake and lower fracture risk. In particular, soy foods and antioxidants from fruit and vegetables are generally considered healthful by consumers [70,71], so they may be associated with a healthier lifestyle and positive habits (e.g., avoiding cigarette smoking, limiting alcohol, exercising regularly) that are typically associated with a lower risk of falls and fractures [72]. In this regard, excessive intake of these compounds should also be avoided, since the recent findings from the InCHIANTI study showed a significant association between risk of hip fracture and higher levels of urinary polyphenols, which represent a reliable markers of polyphenol exposure [73]. 

Our study presents several strengths. First of all, we chose to include only studies in which endpoint data were clearly available. Gray literature and indirect data were excluded in order to prioritize the quality of evidence over numerosity. Second, sensitivity analysis and subgroup analysis were used to investigate the source of heterogeneity. In addition, the quality of the evidence was analyzed using the latest Cochrane tool, which in our opinion appears to be more sensitive than the previous version in identifying potential bias. Nevertheless, some limitations exist. Indeed, since many cases of osteoporosis require drug therapy that could mask the effects of polyphenols, most studies are conducted in healthy women with a BMD or slightly reduced BMD, but others include patients with early forms of osteoporosis, leading to strong heterogeneity in the included population. In addition, the BTM analysis was conducted on a small number of patients and a limited number of markers, since it was a secondary endpoint, so the true effect of polyphenols may have been underestimated. We believe that studies in larger populations with robust methodology and adequate follow-up could be useful in better defining the effect of polyphenols in the prevention and treatment of postmenopausal osteoporosis and strongly recommend that they not be prescribed widely until more robust evidence is available.

## 5. Conclusions

Polyphenols are hormone- and antioxidant-acting substances considered a potential tool for the prevention of postmenopausal osteoporosis. Current clinical evidence does not support the epidemiological data on reduced fracture risk associated with a diet rich in soy derivatives, and suboptimal tolerability, as well as possible gastrointestinal side effects, limits the therapeutic potential of polyphenol supplements in postmenopausal women. Clinicians should be aware that lifestyle modifications with proven efficacy, such as regular physical activity and smoking cessation, should still be recommended even in the presence of a polyphenol-rich diet.

## Figures and Tables

**Figure 1 antioxidants-12-01830-f001:**
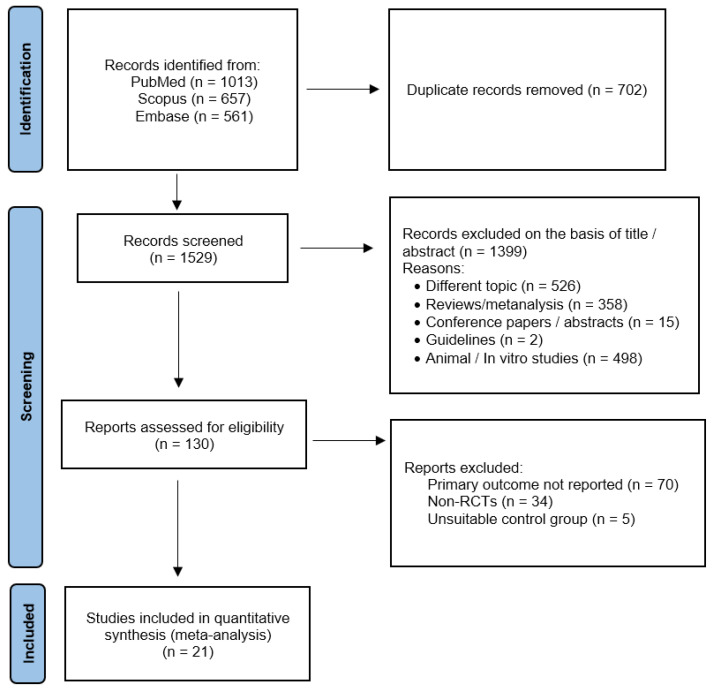
Preferred Reporting Items for Systematic Review and Meta-Analysis Protocols (PRISMA-P) flowchart.

**Figure 2 antioxidants-12-01830-f002:**
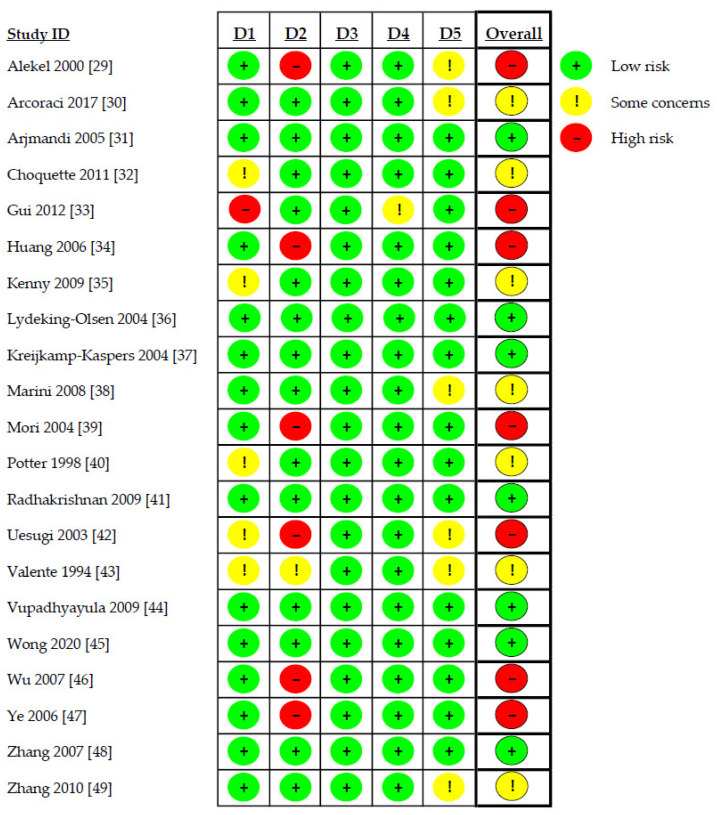
Risk of bias assessment. D1: randomization process; D2: deviations from the intended interventions; D3: missing outcome data; D4: measurement of the outcome; D5: selection of the reported result.

**Figure 3 antioxidants-12-01830-f003:**
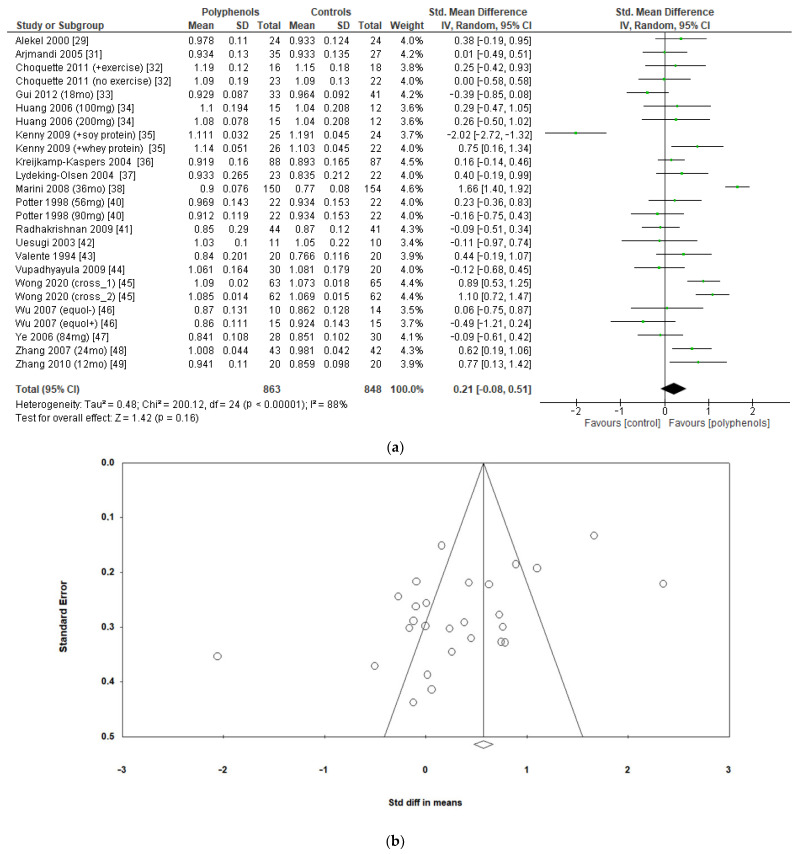
Effects of polyphenols on lumbar bone mineral density: (**a**) forest plot, (**b**) funnel plot.

**Figure 4 antioxidants-12-01830-f004:**
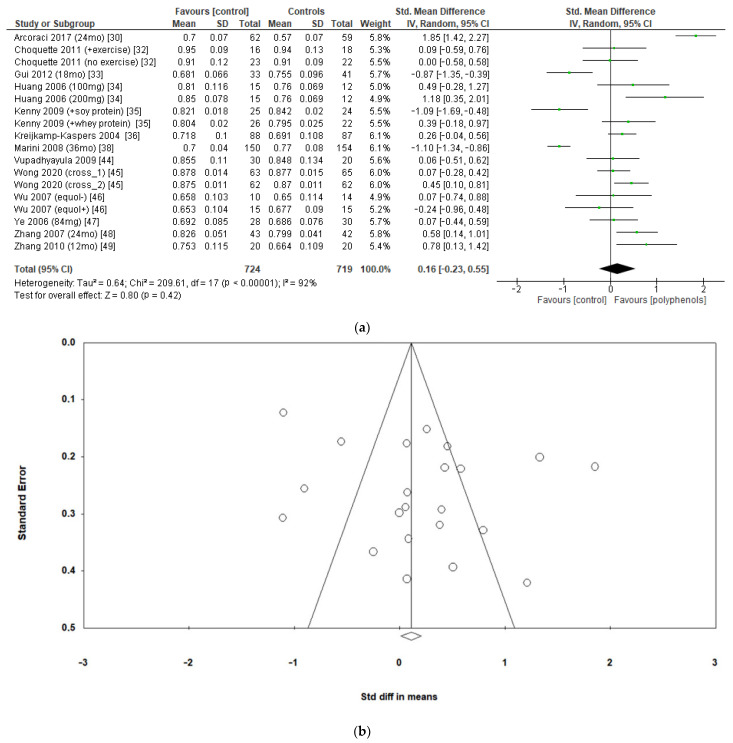
Effects of polyphenols on femoral neck bone mineral density: (**a**) forest plot, (**b**) funnel plot.

**Figure 5 antioxidants-12-01830-f005:**
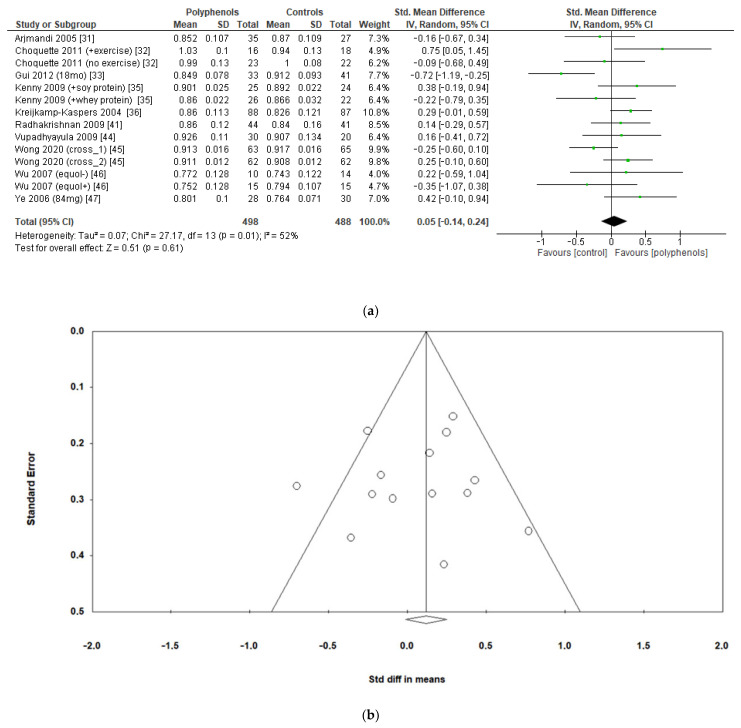
Effects of polyphenols on total hip bone mineral density: (**a**) forest plot, (**b**) funnel plot.

**Figure 6 antioxidants-12-01830-f006:**
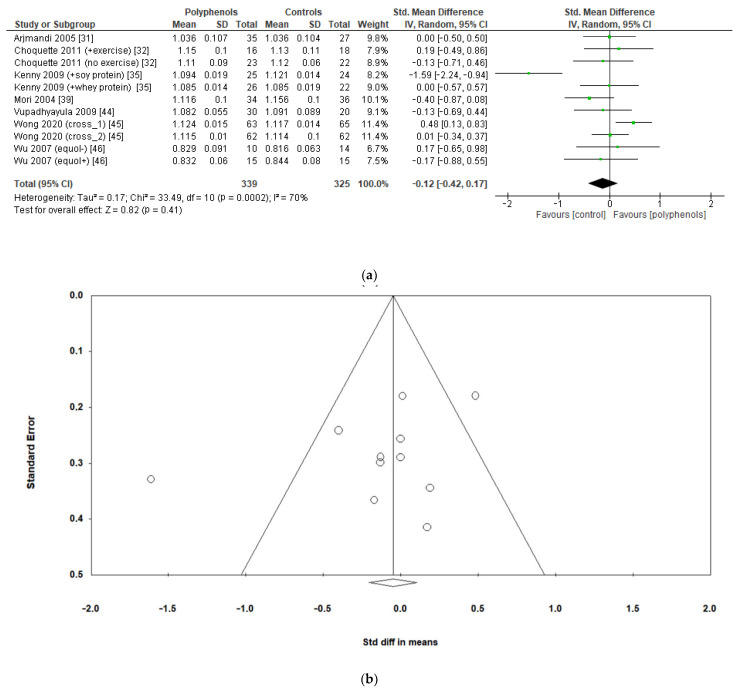
Effects of polyphenols on whole-body BMD bone mineral density: (**a**) forest plot, (**b**) funnel plot.

**Figure 7 antioxidants-12-01830-f007:**
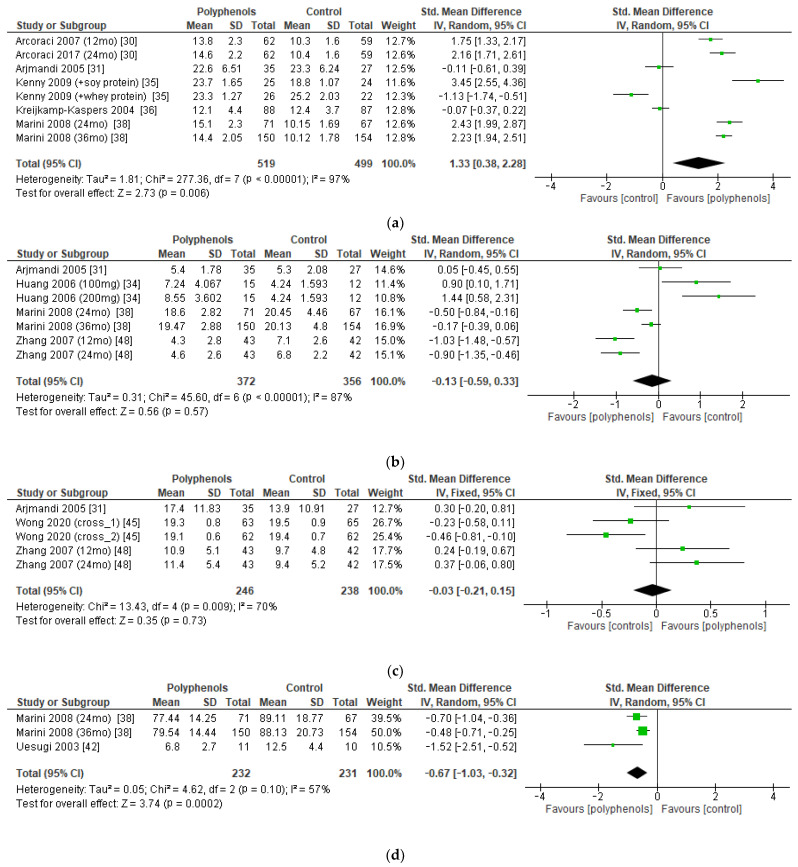
Effects of polyphenols on (**a**) bone-specific alkaline phosphatase, (**b**) deoxypyridinoline, (**c**) osteocalcin, and (**d**) pyridinoline.

**Figure 8 antioxidants-12-01830-f008:**
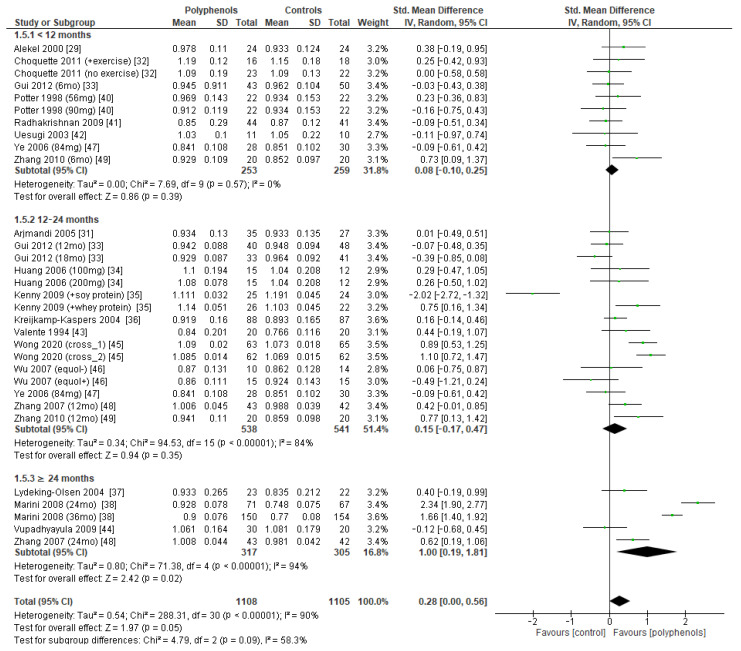
Effects of polyphenols on lumbar bone mineral density according to duration of the study.

**Figure 9 antioxidants-12-01830-f009:**
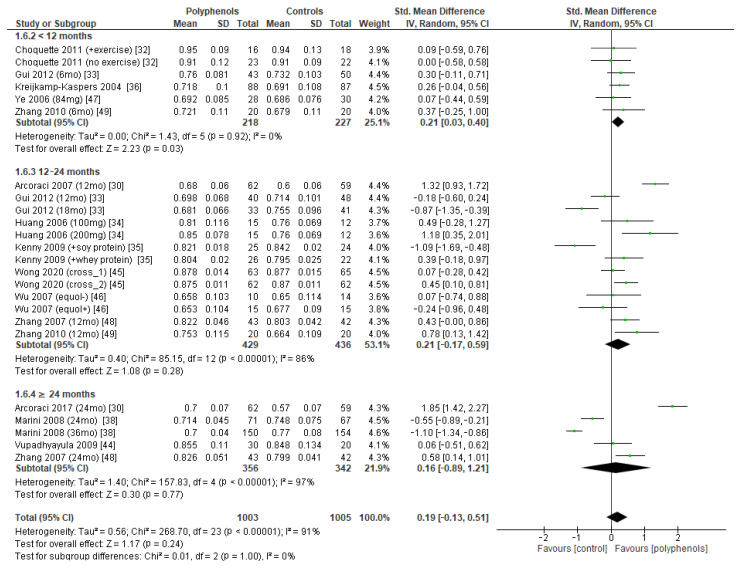
Effects of polyphenols on femoral neck bone mineral density according to duration of the study.

**Figure 10 antioxidants-12-01830-f010:**
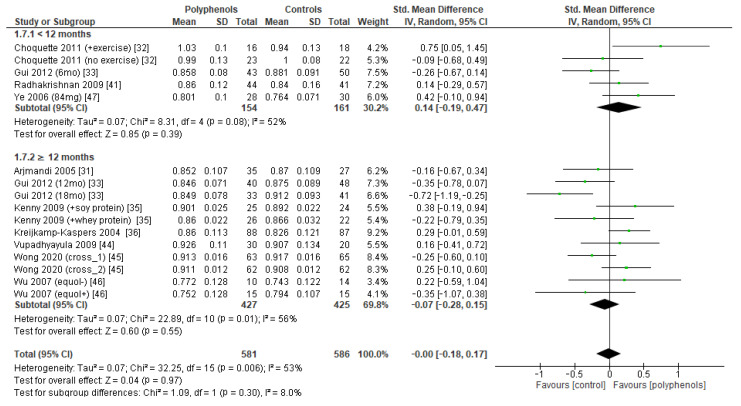
Effects of polyphenols on total hip bone mineral density according to duration of the study.

**Figure 11 antioxidants-12-01830-f011:**
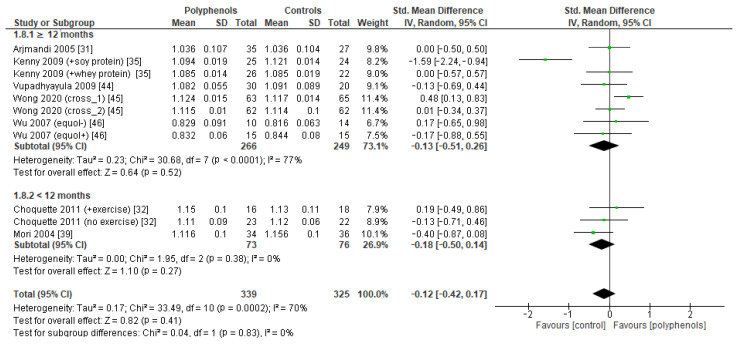
Effects of polyphenols on whole body bone mineral density according to duration of the study.

**Table 1 antioxidants-12-01830-t001:** Characteristics of the included studies at randomization.

First Author	Year	Country	Sample Size	Mean Age ± SD	Type of Polyphenols	Daily Dose (mg)	Additional Therapy	Type of Control
Alekel [29]	2000	USA	69	Poly: 50.2 (median) Ctrl: 50.9 (median)	Soy isoflavones	80.4 (aglycone)	Oral calcium 160 mg/day	Whey protein diet
Arcoraci [30]	2017	Italy	121	Poly: 54.5 ± 2.9 Ctrl: 54.3 ± 2.4	Genistein	54 (aglycone)	1000 mg calcium carbonate and 800 IU vitamin D3	Placebo
Arjmandi [31]	2005	USA	87	Poly: 53 ± 6 Ctrl: 56 ± 6	Soy isoflavones	60 (isoflavones)	25 g proteins from soy products	Regular diet
Choquette [32]	2011	Canada	100	Poly: 61 ± 3 Ctrl: 58 ± 6	Soy isoflavones	70 (isoflavones)	Exercise	Exercise + placebo
Gui [33]	2012	China	100	Poly: 56.1 ± 4.2 Ctrl: 55.8 ± 4.1	Soy isoflavones	3.75–4.5 (isoflavones)	250 mg calcium	Cowmilk
Huang [34]	2006	China	43	Poly: 51.9 ± 5.8 Ctrl: 51.2 ± 1.5	Soy isoflavones	100–200 (isoflavones)	-	Regular diet
Kenny [35]	2009	USA	131	Poly: 73 ± 5.7 Ctrl: 74 ± 6.2	Soy isoflavones	105 (aglycone)	Dietary calcium intake of 1200–1500 mg/d ± soy protein	Placebo + soy protein
Kreijkamp-Kaspers [36]	2004	Netherlands	202	Poly: 66.5 ± 4.7 Ctrl: 66.7 ± 4.8	Soy isoflavones	99 (aglycone)	Riboflavin. Pyridoxine hydrochloride. Cyanocobalamin. Folic acid. Cholecalciferol. And calcium	Placebo
Lydeking-Olsen [37]	2004	Denmark	107	Poly: 57.8 ± 8.4 Ctrl: 56.3 ± 6.7	Soy isoflavones	76 (aglycone)	1500 mg calcium and 200 UI vitamin D	Placebo
Marini [38]	2008	Italy	389	Poly: 53.8 ± 2.9 Ctrl: 53.5 ± 2	Genistein	54 (aglycone)	1000 mg calcium carbonate and 800 IU vitamin D3	Placebo
Mori [39]	2004	Japan	81	Poly: 50.1 ± 4.8 Ctrl: 49.4 ± 4.8	Soy isoflavones	100 (isoflavones)	-	Placebo
Potter [40]	1998	USA	66	Poly: 59 ± 9.1 Ctrl: 61.3 ± 6.3	Soy isoflavones	56–90 (isoflavones)	Calcium phosphate (dosage not available)	Nonfat dry milk
Radhakrishnan [41]	2009	India	100	Poly: 48.07 ± 5.4 Ctrl: 49.71 ± 7.3	Soy isoflavones	75 (isoflavones)	900 mg elemental calcium	Placebo
Uesugi [42]	2003	Japan	22	Poly: 54.9 ± 7.5 Ctrl: 82.5 ± 6.8	Soy isoflavones	61.8 (isoflavones)	-	Placebo
Valente [43]	1994	Italy	40	Poly: 55.9 ± 4.2 Ctrl: 56.8 ± 4.5	Ipriflavone	600 (ipriflavone)	1000 mg calcium	Placebo
Vupadhyayula [44]	2009	USA	203	Poly: 63.42 ± 3.1 Ctrl: 63.63 ± 2.5	Soy isoflavones	90 (aglycone)	500 mg calcium and 125 IU vitamin D	Soy protein without isoflavones
Wong [45]	2020	Australia	146	Poly: 64.3 ± 1.3 Ctrl: 65.8 ± 1.3	Resveratrol	150 (resveratrol)	-	Placebo
Wu [46]	2007	Japan	54	Poly: 54.5 ± 2 Ctrl: 54.8 ± 2.7	Soy isoflavones	75 (isoflavone) 47 (aglycone)	-	Placebo
Ye [47]	2006	China	90	Poly: 52.5 ± 3 Ctrl: 52.7 ± 3.7	Soy isoflavones	84–126 (isoflavones)	-	Placebo
Zhang [48]	2007	China	100	Poly: 64 ± 4 Ctrl: 63 ± 3	EPFs	78 (aglycone)	300 mg elemental calcium	Placebo
Zhang [49]	2010	China	60	NA	Ipriflavone	600 (ipriflavone)	Vitamin AD guttate and 1000 mg compound calcium acid chelate	Placebo and 1000 mg compound calcium acid chelate

Ctrl = controls; EPFs = Epimedium-Derived Phytoestrogen Flavonoids; NA = not available; Poly: polyphenols.

**Table 2 antioxidants-12-01830-t002:** Characteristics of the included studies (follow-up).

First Author	Year	Follow-Up (Months)	Drop-Out	Adherence	DXA Scanner
Alekel [29]	2000	6	0%	Excellent	Hologic QDR2000+
Arcoraci [30]	2017	12–24	NA	Not assessed	Hologic QDR 4500 W
Arjmandi [31]	2005	12	28.7%	Not assessed	Hologic QDR-4500C
Choquette [32]	2011	6	21.0%	Not assessed	Ge Lunar Prodigy
Gui [33]	2012	6–18	14% (6 month) 20% (12 month) 34% (18 month)	Not assessed	Hologic QDR Discovery-W
Huang [34]	2006	12	2.3%	Not assessed	Lunar DPXL
Kenny [35]	2009	12	26.0%	90%	Ge Lunar DPX-IQ
Kreijkamp-Kaspers [36]	2004	12	24.3%	Good	Hologic QDR1000
Lydeking-Olsen [37]	2004	24	11.5%	Not assessed	Norland xR 26 Mark II
Marini [38]	2008	24–36	21.9% (2 year) 64.5% (3 year)	Not assessed	Hologic QDR4500 W
Mori [39]	2004	6	13.6%	Not assessed	GE Lunar DPX-NT
Potter [40]	1998	6	NA	Not assessed	Hologic QDR2000
Radhakrishnan [41]	2009	6	15.0%	Not assessed	Unspecified
Uesugi [42]	2003	3	4.5%	Assessed but not reported	GE Lunar DPX-L
Valente [43]	1994	12	12.5%	Excellent	Hologic QDR 1000
Vupadhyayula [44]	2009	24	58.1%	95%	GE Lunar DPX-L
Wong [45]	2020	12	12.3%	94%	GE Lunar (unspecified)
Wu [46]	2007	12	NA	Not assessed	Hologic QDR-4500A
Ye [47]	2006	6	6.7%	Excellent	Hologic QDR2000+
Zhang [48]	2007	12–24	15.0%	Assessed but not reported	GE Lunar DPX-L
Zhang [49]	2010	6–12	0%	Not assessed	Hologic QDR1000

NA = not available.

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
