# Peer review of "The Effects of Polyphenols on Bone Metabolism in Postmenopausal Women: Systematic Review and Meta-Analysis of Randomized Control Trials"

_antioxidants, 2023, doi:10.3390/antiox12101830_

Round 1

Reviewer 1 Report

The paper regarding the effects of polyphenols on bone metabolism in postmenopausal women: systematic review and meta-analysis of randomized control trials is presented. The authors summarised therapeutic polyphenols on bone metabolism in postmenopausal women. The review is potentially interesting and well-presented.  

There are some possible issues

It seems there is a lack of mechanistic explanation in the discussion on how polyphenols might be beneficial to bone metabolism. Recent studies (for example, PMID: 37653972, PMID: 28325144) have found that phenolic compounds affect osteoclast and ROS signaling as well as osteoblasts which are critical for bone metabolism. It would be informative to discuss these aspects to provide mechanistic insights to promote this paper to the wider readers, in the discussion.

There are some typos.

showing lower incidence of fractures?? Showing a lower incidence of fractures?

we found significant slighy increase?? we found a significantly slight increase?

recommending prescription of polyphenols?? recommending the prescription of polyphenols?

…..

There are some typos.

showing lower incidence of fractures?? Showing a lower incidence of fractures?

we found significant slighy increase?? we found a significantly slight increase?

recommending prescription of polyphenols?? recommending the prescription of polyphenols?

Author Response

The paper regarding the effects of polyphenols on bone metabolism in postmenopausal women: systematic review and meta-analysis of randomized control trials is presented. The authors summarised therapeutic polyphenols on bone metabolism in postmenopausal women. The review is potentially interesting and well-presented.  

A0: We thank the reviewer for his kind comments and constructive criticism. We hope we have resolved all his concerns and thank him for his contribution in improving our work.

There are some possible issues

C1: It seems there is a lack of mechanistic explanation in the discussion on how polyphenols might be beneficial to bone metabolism. Recent studies (for example, PMID: 37653972, PMID: 28325144) have found that phenolic compounds affect osteoclast and ROS signaling as well as osteoblasts which are critical for bone metabolism. It would be informative to discuss these aspects to provide mechanistic insights to promote this paper to the wider readers, in the discussion.

A1: We thank you for this comment. We agree that more space was needed for this topic. Therefore, we have expanded the corresponding section in both the introduction and discussion.

C2: There are some typos.

C2.1: showing lower incidence of fractures?? Showing a lower incidence of fractures?

A2.1: Thanks for your comment. The sentence has been corrected.

C2.2: we found significant slighy increase?? we found a significantly slight increase?

A2.2: Thanks for your comment. The sentence has been corrected.

C2.3: recommending prescription of polyphenols?? recommending the prescription of polyphenols?

A2.3: Thanks for your comment. The sentence has been corrected.

Reviewer 2 Report

The manuscript is well-written and the meta-analysis appears to be properly described and conducted. I think however that the authors have missed many studies in their literature search by not including “isoflavones” as one of their search terms. Please see below for specific comments.

Line 11: When first mentioning polyphenols here, I suggest mentioning a few general food sources that are high in polyphenols.

Line 21: Change “slighy” to “slight”

Lines 31, 35, 42: change “lead” to “leads”

Line 63: Here it is stated polyphenols are high in soy and red clover. It should be mentioned that polyphenols are high in fruits and vegetables.

Line 65: Here where you state polyphenols have high affinity for estrogen receptor beta: Does this refer to all polyphenols or just isoflavones (which are one group of polyphenols)?

Line 84: There are many bone studies with isoflavones, which are a group of polyphenols. I think you should include this in your search terms. I think you have missed a number of studies involving isoflavones in your literature search.

Line 117: superscript the “2” in g/cm2

Line 147: Change “1’739” to 1,739”. The same comment applies to lines 148, 193, 207, 245 and Figure 1

Line 150: It appears that non-English studies were excluded, which introduces bias. Please indicate the number of non-English studies excluded. I suggest attempting to translate and extract data from these articles.

What was the end-date for your literature search?

Figure 8: Why would the “total” result at the bottom of the figure be statistically significant (i.e., combining all studies) but Figure 3b is not statistically significant (which also includes all studies).

I suggest mentioning the statistically significant effect on lumbar spine BMD with treatment of 24 months or greater in the abstract.

The English only requires minor edits

Author Response

The manuscript is well-written and the meta-analysis appears to be properly described and conducted. I think however that the authors have missed many studies in their literature search by not including “isoflavones” as one of their search terms. Please see below for specific comments.

A0:  We sincerely thank the reviewer for taking the time to evaluate our work and for valuable suggestions that certainly improved the original manuscript. The authors agreed with all of the reviewer's concerns and tried to respond appropriately to his comments. See below for specific answers.

C1: Line 11: When first mentioning polyphenols here, I suggest mentioning a few general food sources that are high in polyphenols.

 A1: Thank you for the suggestion. A brief mention has been included in the abstract and the corresponding section in the introduction has been expanded.

C2: Line 21: Change “slighy” to “slight”

A2: Thanks for your comment. The sentence has been corrected.

C3: Lines 31, 35, 42: change “lead” to “leads”

A3: Thanks for your comment. The typos have been corrected.

C4: Line 63: Here it is stated polyphenols are high in soy and red clover. It should be mentioned that polyphenols are high in fruits and vegetables.

A4: We thank you for the suggestion. The sentence has been modified as suggested.

C5: Line 65: Here where you state polyphenols have high affinity for estrogen receptor beta: Does this refer to all polyphenols or just isoflavones (which are one group of polyphenols)?

A5: We agree that the sentence was ambiguous. We have corrected it accordingly.

C6: Line 84: There are many bone studies with isoflavones, which are a group of polyphenols. I think you should include this in your search terms. I think you have missed a number of studies involving isoflavones in your literature search.

A6: We greatly appreciated this comment, which certainly made the methodology of our survey more robust. As suggested by the reviewer, we tried to include the term "isoflavones" in the original search key and saw that about 500 bibliographic entries had been missed during the first search. We then decided to extend search as suggested and found 2 new articles, which were therefore included in our study. Where necessary, all statistical analysis was then repeated and the presentation of data was modified accordingly, including graphs, sensitivity analysis, and estimation of the risk of publication bias.  Unfortunately, there remain many papers on isoflavones that cannot be included because the primary outcome is not reliably presented (e.g., only the T-score without BMD is reported or after treatment the percentage change from baseline is reported instead of the absolute value of BMD).

C7: Line 117: superscript the “2” in g/cm2

A7:  Thanks for your comment. The typo has been corrected.

C8: Line 147: Change “1’739” to 1,739”. The same comment applies to lines 148, 193, 207, 245 and Figure 1

A8: Thanks for your comment. We made all the suggested changes.

C9: Line 150: It appears that non-English studies were excluded, which introduces bias. Please indicate the number of non-English studies excluded. I suggest attempting to translate and extract data from these articles.

A9: We thank the reviewer for this clever comment. In the original manuscript 49 articles were excluded because they were in non-English. In agreement with the reviewer's opinion, the non-English articles were evaluated along with the others, both in terms of title/abstract and as full text, if necessary. Figure 1 was corrected accordingly.

C10: What was the end-date for your literature search?

A10: The literature search was conducted from may to july 2023 (Line 84). All the studies published up to 1 july were included in the screening.

C11: Figure 8: Why would the “total” result at the bottom of the figure be statistically significant (i.e., combining all studies) but Figure 3b is not statistically significant (which also includes all studies).

A11: We thank the reviewer for his keen observation.  The small difference is due to the fact that in the primary outcome analysis, when assessments at different times were available, studies were entered once with the longest follow-up time, as indicated in the statistical analysis section. The model in Figure 8, on the other hand, was created by including all data differentiated by follow-up time, including repeated assessments over the same studies, which would risk overestimating the effect on the primary outcome but is acceptable for subgroup analysis. 

C12: I suggest mentioning the statistically significant effect on lumbar spine BMD with treatment of 24 months or greater in the abstract.

A12: The abstract has been modified as suggested.

Round 2

Reviewer 2 Report

I think the authors have adequately revised their manuscript. I only have a couple of remaining minor suggestions for revision:

Line 16: Change “nineteen” to “twenty-one” since this is the revised number of studies included in the analysis.

Line 79: Change “they” to “the”

The English is fine

Author Response

We sincerely thank again the reviewer for his prompt and appropriate comments. The minor corrections have been made as suggested.